# Consistent Interpolating Ensembles via the Manifold-Hilbert Kernel

**Yutong Wang**
University of Michigan
yutongw@umich.edu

**Clayton D. Scott**
University of Michigan
clayscot@umich.edu

## Abstract

Recent research in the theory of overparametrized learning has sought to establish generalization guarantees in the interpolating regime. Such results have been established for a few common classes of methods, but so far not for ensemble methods. We devise an ensemble classification method that simultaneously interpolates the training data, and is consistent for a broad class of data distributions. To this end, we define the *manifold-Hilbert kernel* for data distributed on a Riemannian manifold. We prove that kernel smoothing regression and classification using the manifold-Hilbert kernel are weakly consistent in the setting of Devroye et al. [22]. For the sphere, we show that the manifold-Hilbert kernel can be realized as a weighted random partition kernel, which arises as an infinite ensemble of partition-based classifiers.

## 1 Introduction

Ensemble methods are among the most often applied learning algorithms, yet their theoretical properties have not been fully understood [12]. Based on empirical evidence, Wyner et al. [42] conjectured that interpolation of the training data plays a key role in explaining the success of AdaBoost and random forests. However, while a few classes of learning methods have been analyzed in the interpolating regime [6, 4], ensembles have not.

Towards developing the theory of interpolating ensembles, we examine an ensemble classification method for data distributed on the sphere, and show that this classifier interpolates the training data and is consistent for a broad class of data distributions. To show this result, we develop two additional contributions that may be of independent interest. First, for data distributed on a Riemannian manifold $M$, we introduce the *manifold-Hilbert kernel* $K_M^{\mathcal{H}}$, a manifold extension of the *Hilbert kernel* [39]. Under the same setting as Devroye et al. [22], we prove that kernel smoothing regression with $K_M^{\mathcal{H}}$ is weakly consistent while interpolating the training data. Consequently, the classifier obtained by taking the sign of the kernel smoothing estimate has zero training error and is consistent.

Second, we introduce a class of kernels called weighted random partition kernels. These are kernels that can be realized as an infinite, weighted ensemble of partition-based histogram classifiers. Our main result is established by showing that when $M = \mathbb{S}^d$, the $d$-dimensional sphere, the manifold-Hilbert kernel is a weighted random partition kernel. In particular, we show that on the sphere, the manifold-Hilbert kernel is a weighted ensemble based on random hyperplane arrangements. This implies that the kernel smoothing classifier is a consistent, interpolating ensemble on $\mathbb{S}^d$. To our knowledge, this is the first demonstration of an interpolating ensemble method that is consistent for a broad class of distributions in arbitrary dimensions.

36th Conference on Neural Information Processing Systems (NeurIPS 2022).

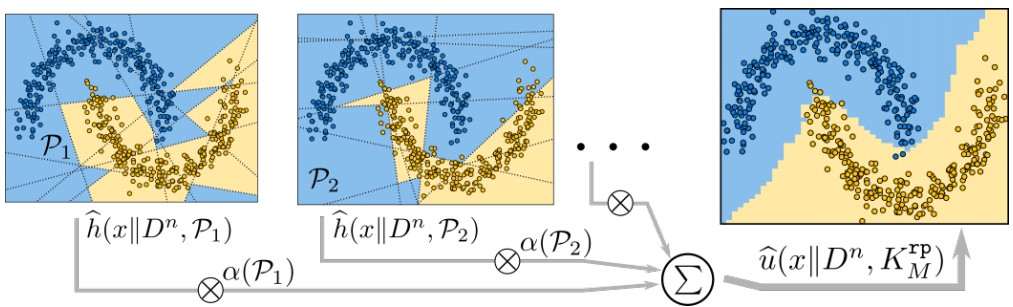

Figure 1: An example of the weighted infinite-ensemble $\widehat{u}(x\|D^n, K_M^{\mathrm{rp}})$ (defined in Eqn. 4) of histogram classifiers $\widehat{h}(x\|D^n, \mathcal{P})$ (defined in Eqn. 2). The partitions $\mathcal{P}_1, \mathcal{P}_2$ are induced by hyperplane arrangements (denoted by dotted lines).

## 1.1 Problem statement

Consider the problem of binary classification on a Riemannian manifold $M$. Let $(X, Y)$ be random variables jointly distributed on $M \times \{\pm 1\}$. Let $D^n := \{(X_i, Y_i)\}_{i=1}^n$ be the (random) training data consisting of $n$ i.i.d copies of $X, Y$. A *classifier*, i.e., a mapping from $D^n$ to a function $\widehat{f}(\bullet \| D^n) : M \to \{\pm 1\}$, has the **interpolating-consistent property** if, when $X$ has a continuous distribution, both of the following hold: 1) $\widehat{f}(X_i \| D^n) = Y_i$, for all $i \in \{1, \dots, n\}$, and 2)

$$\Pr\{\widehat{f}(X\|D^n) \neq Y\} \to \inf_{f:M\to\{\pm 1\} \text{ measurable}} \Pr\{f(X) \neq Y\} \quad \text{in probability as } n \to \infty. \quad (1)$$

Our goal is to find an interpolating-consistent ensemble of *histogram classifiers*, to be defined below.

A *partition* on $M$, denoted by $\mathcal{P}$, is a set of subsets of $M$ such that $P \cap P' = \emptyset$ for all $P, P' \in \mathcal{P}$ and $M = \bigcup_{P \in \mathcal{P}} P$. Given $x \in M$, let $\mathcal{P}[x]$ denote the unique element $P \in \mathcal{P}$ such that $x \in P$. The set of all partitions on a space $M$ is denoted $\mathtt{Part}(M)$. The *histogram classifier* with respect to $D^n$ over $\mathcal{P}$ is the sign of the function $\widehat{h}(\bullet \| D^n, \mathcal{P}) : M \to \mathbb{R}$ given by

$$\widehat{h}(x\|D^n, \mathcal{P}) := \sum_{i=1}^n Y_i \cdot \mathbb{I}\{x \in \mathcal{P}[X_i]\}, \quad (2)$$

where $\mathbb{I}$ is the indicator function. See Figure 1-left panels.

**Definition 1.1.** A *weighted random partition* (WRP) over $M$ is a 3-tuple $(\Theta, \mathfrak{P}, \alpha)$ consisting of (i) *parameter space of partitions*: a set $\Theta$ where $\mathcal{P}_\theta \in \mathtt{Part}(M)$ for each $\theta \in \Theta$, (ii) *random partitions*: a probability measure $\mathfrak{P}$ on $\Theta$, and (iii) *weights*: a nonnegative function $\alpha : \Theta \to \mathbb{R}_{\geq 0}$ that is integrable with respect to the measure $\mathfrak{P}$.

**Example 1.2** (Regular partition of the $d$-cube). Let $M = [0,1]^d$ and $\Theta = \{1, 2 \dots\} =: \mathbb{N}_+$. For each $n \in \mathbb{N}_+$, denote by $\mathcal{P}_n$ the regular partition of $M$ into $n^d$ $d$-cubes of side length $1/n$. For any probability mass function $\mathfrak{P}$ on $\mathbb{N}_+$ and weights $\alpha : \mathbb{N}_+ \to \mathbb{R}_{\geq 0}$, the 3-tuple $(\Theta, \mathfrak{P}, \alpha)$ is a WRP.

Below, WRPs will be denoted with 2-letter names in the sans-serif font, e.g., "rp" for a generic WRP, and "ha" for the weighted hyperplane arrangement random partition (Definition 5.1). The *weighted random partition kernel* associated to $\mathsf{rp} = (\Theta, \mathfrak{P}, \alpha)$ is defined as

$$K_M^{\mathrm{rp}} : M \times M \to \mathbb{R}_{\geq 0} \cup \{\infty\}, \quad K_M^{\mathrm{rp}}(x, z) := \mathbb{E}_{\theta \sim \mathfrak{P}}[\alpha(\theta)\mathbb{I}\{x \in \mathcal{P}_\theta[z]\}]. \quad (3)$$

When $\alpha \equiv 1$, we recover the notion of unweighted random partition kernel introduced in [21]. Note that the kernel is symmetric since $\mathbb{I}\{x \in \mathcal{P}_\theta[z]\} = \mathbb{I}\{z \in \mathcal{P}_\theta[x]\}$. If $K_M^{\mathrm{rp}} < \infty$, then $K_M^{\mathrm{rp}}$ is a positive definite (PD) kernel. When $K_M^{\mathrm{rp}}$ can evaluate to $\infty$, the definition of a PD kernel is not applicable since the positive definite property is defined only for to kernels taking finite values [10].

Let $\mathtt{sgn} : \mathbb{R} \cup \{\pm\infty\} \to \{\pm 1\}$ be the sign function. For a WRP, define the weighted infinite-ensemble

$$\widehat{u}(x\|D^n, K_M^{\mathrm{rp}}) := \sum_{i=1}^n Y_i \cdot K_M^{\mathrm{rp}}(x, X_i) = \mathbb{E}_{\theta \sim \mathfrak{P}}[\alpha(\theta)\widehat{h}(x\|D^n, \mathcal{P}_\theta)]. \quad (4)$$

Note that the equality on the right follows immediately from linearity of the expectation and the definition of $\widehat{h}(\bullet \| D^n, \mathcal{P}_\theta)$ in Equation (2). See Figure 1-right panel.

**Main problem.** Find a WRP such that $\mathtt{sgn}(\widehat{u}(\bullet \| D^n, K_M^{\mathrm{rp}}))$ has the interpolating-consistent property.

## 1.2 Outline of approach and contributions

In the regression setting, we have $(X, Y)$ jointly distributed on $M \times \mathbb{R}$. Let $m(x) := \mathbb{E}[Y|X = x]$. Recall from Belkin et al. [8, Equation (7)] the definition of the *kernel smoothing estimator* with a so-called *singular*[1] kernel $K : M \times M \to [0, +\infty]$:

$$\widehat{m}(x \| D^n, K) := \begin{cases} Y_i & : \exists i \in [n] \text{ such that } x = X_i \\ \frac{\sum_{i=1}^n Y_i K(x, X_i)}{\sum_{j=1}^n K(x, X_j)} & : \sum_{j=1}^n K(x, X_j) > 0 \\ 0 & : \text{otherwise.} \end{cases} \tag{5}$$

We note that Equation (5) is referred as the *Nadaraya-Watson* estimate in [8]. Now, we simply write $\widehat{m}_n(x)$ instead of $\widehat{m}(x \| D^n, K)$ when there is no ambiguity. Similarly, we write $\widehat{u}_n(x)$ instead of $\widehat{u}(x \| D^n, K)$ from earlier. Note that $\mathtt{sgn}(\widehat{m}_n(x)) = \mathtt{sgn}(\widehat{u}_n(x))$ if $\sum_{j=1}^n K(x, X_j) > 0$.

Observe that $\widehat{m}_n$ is interpolating by construction. Let $\mu_X$ denote the marginal distribution of $X$. The $L_1$-*error* of $\widehat{m}_n$ in approximating $m$ is $J_n := \int_M |\widehat{m}_n(x) - m(x)| \mu_X(dx)$. For $M = \mathbb{R}^d$ and the *Hilbert kernel* defined by $K_{\mathbb{R}^d}^{\mathcal{H}}(x, z) := \|x - z\|^{-d}$, Devroye et al. [22] proved $L_1$-consistency for *regression*: $J_n \to 0$ in probability when $Y$ is bounded and $X$ is continuously distributed.

**Our contributions.** Our primary contribution is to demonstrate an ensemble method with the consistent-interpolating property. Toward this end, in Section 3, we introduce the manifold-Hilbert kernel $K_M^{\mathcal{H}}$ on a Riemannian manifold $M$. When show that when $M$ is complete, connected, and smooth, kernel smoothing regression with $K_M^{\mathcal{H}}$ has the same consistency guarantee (Theorem 3.2) as $K_{\mathbb{R}^d}^{\mathcal{H}}$ mentioned in the preceding paragraph. In Section 5, we consider the case when $M = \mathbb{S}^d$, and show that the manifold-Hilbert kernel $K_{\mathbb{S}^d}^{\mathcal{H}}$ is a weighted random partition kernel (Proposition 5.2).

Devroye et al. [22, Section 7] observed that the $L_1$-consistency of $\widehat{m}_n$ for regression implies the consistency for classification of $\mathtt{sgn} \circ \widehat{u}_n$. Furthermore, $\widehat{m}_n$ is interpolating for regression implies that $\mathtt{sgn} \circ \widehat{u}_n$ is interpolating for classification. These observations together with our results demonstrate the existence of a weighted infinite-ensemble classifier with the interpolating-consistent property.

## 1.3 Related work

**Kernel regression.** Kernel smoothing regression, or simply kernel regression, is an interpolator when the kernel used is singular, a fact known to Shepard [39] in 1968. Devroye et al. [22] showed that kernel regression with the Hilbert kernel is interpolating and weakly consistent for data with a density and bounded labels. Using singular kernels with compact support, Belkin et al. [8] showed that minimax optimality can be achieved under additional distributional assumptions.

**Random forests.** Wyner et al. [42] proposed that interpolation may be a key mechanism for the success of random forests and gave a compelling intuitive rationale. Belkin et al. [6] studied empirically the double descent phenomenon in random forests by considering the generalization performance past the interpolation threshold. The PERT variant of random forests, introduced by Cutler and Zhao [20], provably interpolates in 1-dimension. Belkin et al. [7] pose as an interesting question whether the result of Cutler and Zhao [20] extends to higher dimension. Many work have established consistency of random forest and its variants under different settings [15, 11, 38]. However, none of these work addressed interpolation.

**Boosting.** For classification under the noiseless setting (i.e., the Bayes error is zero), AdaBoost is interpolating and consistent (see Freund and Schapire [26, first paragraph of Chapter 12]). However, this setting is too restrictive and the result does not answer if consistency is possible when fitting the noise. Bartlett and Traskin [5] proved that AdaBoost with early stopping is universally consistent, however without the interpolation guarantee. To the best of our knowledge, whether AdaBoost or any other variant of boosting can be interpolating and consistent remains open.

**Random partition kernels.** Breiman [16] and Geurts et al. [28] studied infinite ensembles of simplified variants of random forest and connections to certain kernels. Davies and Ghahramani [21] formalized this connection and coined the term *random partition kernel*. Scornet [37] further developed the theory of random forest kernels and obtained upper bounds on the rate of convergence. However, it is not clear if these variants of random forests are interpolating.

---

[1]The "singular" modifier refers to the fact that $K(x, x) = +\infty$ for all $x \in M$.

Previously defined (unweighted) random partition kernels are bounded, and thus cannot be singular. On the other hand, the manifold-Hilbert kernel is always singular. To bridge between ensemble methods and theory on interpolating kernel smoothing regression, we propose *weighted* random partitions (Definition 1.1), whose associated kernel (Equation 3) can be singular.

**Learning on Riemannian manifolds.** Strong consistency of a kernel-based classification method on manifolds has been established by Loubes and Pelletier [32]. However, the result requires the kernel to be bounded and thus the method is not guaranteed to be interpolating. See Feragen and Hauberg [25] for a review of theoretical results regarding kernels on Riemannian manifolds.

Beyond kernel methods, other classical methods for Euclidean data have been extended to Riemannian manifolds, e.g., regression [40], classification [43], and dimensionality reduction and clustering [44][34]. To the best of our knowledge, no previous works have demonstrated an interpolating-consistent classifiers on manifolds other than $\mathbb{R}^d$.

In many applications, the data naturally belong to a Riemannian manifold. Spherical data arise from a range of disciplines in natural sciences. See the influential textbook by Mardia and Jupp [33, Ch.1§4]. For applications of the Grassmanian manifold in computer vision, see Jayasumana et al. [30] and the references therein. Topological data analysis [41] presents another interesting setting of manifold-valued data in the form of *persistence diagrams* [3, 31].

## 2   Background on Riemannian Manifolds

We give an intuitive overview of the necessary concepts and results on Riemannian manifolds. A longer, more precise version of this overview is in the Supplemental Materials Section A.1.

A smooth $d$-dimensional manifold $M$ is a topological space that is locally diffeomorphic[2] to open subsets of $\mathbb{R}^d$. For simplicity, suppose that $M$ is embedded in $\mathbb{R}^N$ for some $N \geq d$, e.g., $\mathbb{S}^d \subseteq \mathbb{R}^{d+1}$. Let $x \in M$ be a point. The *tangent space* at $x$, denoted $T_x M$, is the set of vectors that is tangent to $M$ at $x$. Since linear combinations of tangent vectors are also tangent, the tangent space $T_x M$ is a vector space. Tangent vectors can also be viewed as the time derivative of smooth curves. In particular, let $x \in M$. If $\epsilon > 0$ is an open set and $\gamma : (-\epsilon, \epsilon) \to M$ is a smooth curve such that $\gamma(0) = x$, then $\frac{d\gamma}{dt}(0) \in T_x M$.

A *Riemannian metric* on $M$ is a choice of inner product $\langle \cdot, \cdot \rangle_x$ on $T_x M$ for each $x$ such that $\langle \cdot, \cdot \rangle_x$ varies smoothly with $x$. Naturally, $\|z\|_x := \sqrt{\langle z, z \rangle_x}$ defines a norm on $T_x M$. The length of a piecewise smooth curve $\gamma : [a, b] \to M$ is defined by $\texttt{len}(\gamma) := \int_a^b \|\dot{\gamma}(t)\|_{\gamma(t)} dt$. Define $\texttt{dist}_M(x, \xi) := \inf\{\texttt{len}(\gamma) : \gamma \text{ is a piecewise smooth curve from } x \text{ to } \xi\}$, which is a metric on $M$ in the sense of metric spaces (see Sakai [36, Proposition 1.1]). For $x \in M$ and $r \in (0, \infty)$, the open metric ball centered at $x$ of radius $r$ is denoted $\texttt{B}_x(r, M) := \{\xi \in M : \texttt{dist}_M(x, \xi) < r\}$.

A curve $\gamma : [a, b] \to M$ is a geodesic if $\gamma$ is *locally* distance minimizing and has constant speed, i.e., $\|\frac{d\gamma}{dt}(\tau)\|_{\gamma(\tau)}$ is constant. Now, suppose $x \in M$ and $v \in T_x M$ are such that there exists a geodesic $\gamma : [0, 1] \to M$ where $\gamma(0) = x$ and $\frac{d\gamma}{dt}(0) = v$. Define $\exp_x(v) := \gamma(1)$, the element reached by traveling along $\gamma$ at time $= 1$. See Figure 2 for the case when $M = \mathbb{S}^2$.

For a fixed $x \in M$, the above function $\exp_x$, the *exponential map*, can be defined on an open subset of $T_x M$ containing the origin. The Hopf-Rinow theorem ([23, Ch. 8, Theorem 2.8]) states that if $M$ is connected and complete with respect to the metric $\texttt{dist}_M$, then $\exp_x$ can be defined on all of $T_x M$.

## 3   The Manifold-Hilbert kernel

Throughout the remainder of this work, we assume that $M$ is a complete, connected, and smooth Riemannian manifold of dimension $d$.

**Definition 3.1.** We define the *manifold-Hilbert kernel* $K_M^{\mathcal{H}} : M \times M \to [0, \infty]$ for each $x, \xi \in M$ by $K_M^{\mathcal{H}}(x, \xi) := \texttt{dist}_M(x, \xi)^{-d}$ if $x \neq \xi$ and $K_M^{\mathcal{H}}(x, x) := \infty$ otherwise.

---

[2]A diffeomorphism is a smooth bijection whose inverse is also smooth.

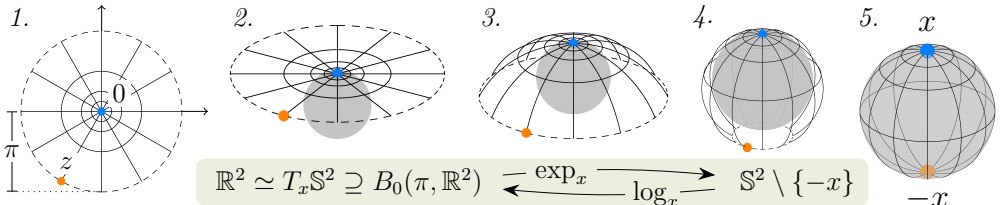

Figure 2: An illustration of the exponential map $\exp_x$ for the manifold $M = \mathbb{S}^2$, where $x$ is the "northpole" (blue) and $-x$ the "southpole" (orange). The logarithm map $\log_x$, discussed in Section 4.1, is a right-inverse to $\exp_x$, i.e., $\exp_x \circ \log_x$ is the identity. *Panel 1.* The tangent space $T_x\mathbb{S}^2$ visualized as $\mathbb{R}^2$. The dashed circle encloses a disc of radius $\pi$. *Panel 2.* The tangent space realized as the hyperplane tangent to sphere at $x$. *Panels 3-5.* Animation showing $\exp_x$ as a bijection from the open disc of radius $\pi$ to $\mathbb{S}^2 \setminus \{-x\}$. The entire dashed circle in Panel i is mapped to $-x$ the southpole. Thus, $\log_x$ maps the southpole $-x$ to a point $z$ on the dashed circle.

Let $\lambda_M$ be the *Riemann–Lebesgue volume measure* of $M$. Integration with respect to this measure is denoted $\int_M f d\lambda_M$ for a function $f : M \to \mathbb{R}$. For details of the construction of $\lambda_M$, see Amann and Escher [1, Proposition 1.5]. When $M = \mathbb{R}^d$, $\lambda_M$ is the ordinary Lebesgue measure and $\int_{\mathbb{R}^d} f d\lambda_{\mathbb{R}^d}$ is the ordinary Lebesgue integral. For this case, we simply write $\lambda$ instead of $\lambda_{\mathbb{R}^d}$.

We now state our first main result, a manifold theory extension of Devroye et al. [22, Theorem 1].

**Theorem 3.2.** *Suppose that $X$ has a density $f_X$ with respect to $\lambda_M$ and that $Y$ is bounded. Let $P_{Y|X}$ be a conditional distribution of $Y$ given $X$ and $m_{Y|X}$ be its conditional expectation. Let $\widehat{m}_n(x) := \widehat{m}(x \| D^n, K_M^{\mathcal{H}})$. Then*

1. *at almost all $x \in M$ with $f_X(x) > 0$, we have $\widehat{m}_n(x) \to m_{Y|X}(x)$ in probability,*
2. *$J_n := \int_M |\widehat{m}_n(x) - m_{Y|X}(x)| f_X(x) d\lambda_M(x) \to 0$ in probability.*

In words, the kernel smoothing regression estimate $\widehat{m}_n$ based on the manifold-Hilbert kernel is consistent and interpolates the training data, provided $X$ has a density and $Y$ is bounded. As a consequence, following the same logic as in Devroye et al. [22], the associated classifier $\mathrm{sgn} \circ \widehat{u}_n$ has the interpolating-consistent property. Before proving Theorem 3.2, we first review key concepts in probability theory on Riemannian manifolds.

## 3.1 Probability on Riemannian manifolds

Let $\mathcal{B}_M$ be the Borel $\sigma$-algebra of $M$, i.e., the smallest $\sigma$-algebra containing all open subsets of $M$. We recall the definition of $M$-valued random variables, following Pennec [35, Definition 2]:

**Definition 3.3.** Let $(\Omega, \mathbb{P}, \mathcal{A})$ be a probability space with measure $\mathbb{P}$ and $\sigma$-algebra $\mathcal{A}$. A $M$-*valued random variable $X$* is a Borel-measurable function $\Omega \to M$, i.e., $X^{-1}(B) \in \mathcal{A}$ for all $B \in \mathcal{B}_M$.

**Definition 3.4** (Density). A random variable $X$ taking values in $M$ has a *density* if there exists a nonnegative Borel-measurable function $f : M \to [0, \infty]$ such that for all Borel sets $B$ in $M$, we have $\Pr(X \in B) = \int_B f d\lambda_M$. The function $f$ is said to be a *probability density function* (PDF) of $X$.

Next, we recall the definition of conditional distributions, following Dudley [24, Ch. 10 §2]:

**Definition 3.5** (Conditional distribution[3]). Let $(X, Y)$ be a random variable jointly distributed on $M \times \mathbb{R}$. Let $P_X(\cdot)$ be the probability measure corresponding to the marginal distribution of $X$. A *conditional distribution* for $Y$ given $X$ is a collection of probability measures $P_{Y|X}(\cdot|x)$ on $\mathbb{R}$ indexed by $x \in M$ satisfying the following:

1. For all Borel sets $A \subseteq \mathbb{R}$, the function $M \ni x \mapsto P_{Y|X}(A|x) \in [0, 1]$ is Borel-measurable.
2. For all $A \subseteq \mathbb{R}$ and $B \subseteq M$ Borel sets, $\Pr(Y \in A, X \in B) = \int_B P_{Y|X}(A|x) P_X(dx)$.

The *conditional expectation*[4] is defined as $m_{Y|X}(x) := \int_{\mathbb{R}} y P_{Y|X}(dy|x)$.

---

[3] also known as *disintegration measures* according to Chang and Pollard [18].

[4] More often, the conditional expectation is denoted $\mathbb{E}[Y|X = x]$. However, our notation is more convenient for function composition and compatible with that of [22].

The existence of a conditional probability for a joint distribution $(X, Y)$ is guaranteed by Dudley [24, Theorem 10.2.2]. When $(X, Y)$ has a joint density $f_{XY}$ and marginal density $f_X$, the above definition gives the classical formula $P_{Y|X}(A|x) = \int_A f_{XY}(x, y)/f_X(x)dy$ when $\infty > f_X(x) > 0$. See the first example in Dudley [24, Ch. 10 §2].

## 3.2 Lebesgue points on manifolds

Devroye et al. [22] proved Theorem 3.2 when $M = \mathbb{R}^d$ and, moreover, that part 1 holds for the so-called *Lebesgue points*, whose definition we now recall.

**Definition 3.6.** Let $f : M \to \mathbb{R}$ be an absolutely integrable function and $x \in M$. We say that $x$ is a *Lebesgue point* of $f$ if $f(x) = \lim_{r \to 0} \frac{1}{\lambda_M(\text{B}_x(r, M))} \int_{\text{B}_x(r, M)} f d\lambda_M$.

For an integrable function, the following result states that almost all points are its Lebesgue points. For the proof, see Fukuoka [27, Remark 2.4].

**Theorem 3.7** (Lebesgue differentation). *Let $f : M \to \mathbb{R}$ be an absolutely integrable function. Then there exists a set $A \subseteq M$ such that $\lambda_M(A) = 0$ and every $x \in M \setminus A$ is a Lebesgue point of $f$.*

Next, for the reader's convenience, we restate Devroye et al. [22, Theorem 1], emphasizing the connection to Lebesgue points. The result will be used in our proof of Theorem 3.2 in the next section.

**Theorem 3.8** (Devroye et al. [22]). *Let $M = \mathbb{R}^d$ be the flat Euclidean space. Then Theorem 3.2 holds. Moreover, Part 1 holds for all $x$ that is a Lebesgue point to both $f_X$ and $m_{Y|X} \cdot f_X$.*

# 4 Proof of Theorem 3.2

The focal point of the first subsection is Lemma 4.1 which shows the Borel measurability of extensions of the so-called Riemannian logarithm. The second subsection contains two key results regarding densities of $M$-valued random variables transformed by the Riemannian logarithm. The final subsection proves Theorem 3.2 leveraging results from the preceding two subsections.

## 4.1 The Riemannian logarithm

Throughout, $x$ is assumed to be an arbitrary point of $M$. Let $U_x M = \{v \in T_x M : \|v\|_x = 1\} \subseteq T_x M$ denote the set of unit tangent vectors. Define a function $\tau_x : U_x M \to (0, \infty]$ as follows[5]:

$$\tau_x(u) := \sup\{t > 0 : t = \text{dist}_M(x, \exp_x(tu))\}.$$

The *tangent cut locus* is the set $\tilde{C}_x \subseteq T_x M$ defined by $\tilde{C}_x := \{\tau_x(u)u : u \in U_x M, \tau_x(u) < \infty\}$. Note that it is possible for $\tau_x(u) = \infty$ for all $u \in U_x M$ in which case $\tilde{C}_x$ is empty. The *cut locus* is the set $C_x := \exp_x(\tilde{C}_x) \subseteq M$.

The *tangent interior set* is $\tilde{I}_x := \{tu : 0 \le t < \tau_x(u), u \in U_x M\}$ and the *interior set* is the set $I_x := \exp_x(\tilde{I}_x)$. Finally, define $\tilde{D}_x := \tilde{I}_x \cup \tilde{C}_x$. Note that for each $z = tu \in \tilde{I}_x$, we have

$$\|z\|_x = t = \text{dist}_M(x, \exp_x(tu)) = \text{dist}_M(x, \exp_x(z)). \tag{6}$$

Consider the example where $M = \mathbb{S}^2$ as in Figure 2. Then $\tau_x(u) = \pi$ for all $u \in U_x M$. Thus, the tangent interior set $\tilde{I}_x = \text{B}_0(\pi, \mathbb{R}^2)$, the open disc of radius $\pi$ centered at the origin.

When restricted to $\tilde{I}_x$, the exponential map $\exp_x |_{\tilde{I}_x} : \tilde{I}_x \to I_x$ is a diffeomorphism. Its functional inverse, denoted by $\log_x |_{I_x}$, is called the *Riemannian Logarithm* [9, 45]. In previous works, $\log_x |_{I_x}$ is only defined from $I_x$ to $\tilde{I}_x$. The next result shows that the domain of $\log_x |_{I_x} : I_x \to \tilde{I}_x$ can be extended to $\log_x : M \to \tilde{D}_x$ while remaining Borel-measurable.

**Lemma 4.1.** *For all $x \in M$, there exists a Borel measurable map $\log_x : M \to T_x M$ such that $\log_x(M) \subseteq \tilde{D}_x$ and $\exp_x \circ \log_x$ is the identity on $M$. Furthermore, for all $x, \xi \in M$, we have $\text{dist}_M(x, \xi) = \|\log_x(\xi)\|_x$.*

---

[5]Positivity of $\tau_x$ is asserted at Sakai [36, eq. (4.1)]

*Proof sketch.* The full proof of the lemma is provided in Section A.2 of the Supplemental Materials[6]. Below, we illustrate the idea of the proof using the example when $M = \mathbb{S}^2$ as in Figure 2.

Let $x \in \mathbb{S}^2$ be the "northpole" (the blue point). The tangent cut locus $\tilde{C}_x$ is the dashed circle in the left panel of Figure 2. The exponential map $\exp_x$ is one-to-one on $\tilde{D}_x$ except on the dashed circle, which all gets mapped to $-x$, the "southpole" (the orange point). A consequence of the measurable selection theorem[7] is that $\log_x$ can be extended to be a Borel-measurable right inverse of $\exp_x$ by selecting $z$ point on $\tilde{C}_x$ such that $\log_x(-x) = z$. $\qquad\square$

Thus, we've shown that $\log_x : M \to T_x M$ is Borel-measurable. Now, recall that $T_x M$ is equipped with the inner product $\langle \cdot, \cdot \rangle_x$, i.e., the Riemannian metric. Below, for each $x \in M$ choose an orthonormal basis on $T_x M$ with respect to $\langle \cdot, \cdot \rangle$. Then $T_x M$ is isomorphic as an inner product space to $\mathbb{R}^d$ with the usual dot product.

Our next two results are "change-of-variables formulas" for computing the densities/conditional distributions of $M$-valued random variables after the $\log_x$ transform. Recall that $\lambda_M$ is the Riemann-Lebesgue measure on $M$ and $\lambda$ is the ordinary Lebesgue measure on $\mathbb{R}^d = T_x M$.

**Proposition 4.2.** *Let $x \in M$ be fixed. There exists a Borel measurable function $\nu_x : M \to \mathbb{R}$ with the following properties:*

(i) *Let $X$ be a random variable on $M$ with density $f_X$ and let $Z := \log_x(X)$. Then $Z$ is a random variable on $T_x M$ with density $f_Z(z) := f_X(\exp_x(z)) \cdot \nu_x(\exp_x(z))$.*
(ii) *Let $f : M \to \mathbb{R}$ be an absolutely integrable function such that $x$ is a Lebesgue point of $f$. Define $f : T_x M \to \mathbb{R}$ by $h(z) := f(\exp_x(z)) \cdot \nu_x(\exp_x(z))$. Then $0 \in T_x M$ is a Lebesgue point for $h$.*

**Proposition 4.3.** *Let $(X, Y)$ have a joint distribution on $M \times \mathbb{R}$ such that the marginal of $X$ has a density $f_X$ on $M$. Let $P_{Y|X}(\cdot|\cdot)$ be a conditional distribution for $Y$ given $X$. Let $x \in M$. Define $Z := \log_x(X)$ and consider the joint distribution $(Z, Y)$ on $T_p M \times \mathbb{R}$. Then $P_{Y|Z}(\cdot|\cdot) := P_{Y|X}(\cdot| \exp_x(\cdot))$ is a conditional distribution for $Y$ given $Z$. Consequently, $m_{Y|X} \circ \exp_x = m_{Y|Z}$.*

The above propositions are straightforward manifold-theoretic extensions of well-known results on Euclidean spaces. For completeness, the full proofs are in Supplemental Materials Section A.4. An anonymous reviewer brought to our attention that Proposition 4.2 is the consequence of a well-known formula from geometric measure theory, called the area formula [2, p. 44-45].

## 4.2 Proof of Theorem 3.2

Fix $x \in M$ such that $x$ is a Lebesgue point of $f_X$ and $m_{Y|X} \cdot f_X$. Note that by Theorem 3.7, almost all $x \in M$ has this property. Next, let $Z = \log_x(X)$ and $f_Z$ be as in Proposition 4.2-(i). Then

1. $f_Z = (f_X \circ \exp_x) \cdot (\nu_x \circ \exp_x)$, and
2. $(m_{Y|X} \circ \exp_x) \cdot f_Z = (m_{Y|X} \circ \exp_x) \cdot (f_X \circ \exp_x) \cdot (\nu_x \circ \exp_x)$.

Now, proposition 4.2-(ii) implies that $0$ is a Lebesgue point of both $f_Z$ and $(m_{Y|X} \circ \exp_x) \cdot f_Z$. Furthermore, by Proposition 4.3, we have $m_{Y|X} \circ \exp_x = m_{Y|Z}$. Thus, $0$ is a Lebesgue point of $f_Z$ and $m_{Y|Z} \cdot f_Z$.

Now, let $D_n := \{(X_i, Y_i)\}_{i \in [n]}$. Define $Z_i := \log_x(X_i)$, which are i.i.d copies of the random variable $Z := \log_x(X)$, and let $\tilde{D}_n := \{(Z_i, Y_i)\}_{i \in [n]}$. Then we have

$$\widehat{m}(x\|D^n, K_M^{\mathcal{H}}) \overset{(a)}{=} \frac{\sum_{i=1}^n Y_i \cdot \operatorname{dist}_M(x, X_i)^{-d}}{\sum_{j=1}^n \operatorname{dist}_M(x, X_j)^{-d}} \overset{(b)}{=} \frac{\sum_{i=1}^n Y_i \cdot \|Z_i\|_x^{-d}}{\sum_{j=1}^n \|Z_j\|_x^{-d}}$$

$$\overset{(c)}{=} \frac{\sum_{i=1}^n Y_i \cdot \operatorname{dist}_{\mathbb{R}^d}(0, Z_i)^{-d}}{\sum_{j=1}^n \operatorname{dist}_{\mathbb{R}^d}(0, Z_j)^{-d}} \overset{(d)}{=} \widehat{m}(0\|\tilde{D}^n, K_{\mathbb{R}^d}^{\mathcal{H}})$$

---

[6] An anonymous reviewer has provided a shorter, alternative proof of Lemma 4.1. See `https://openreview.net/forum?id=zqQKGaNI4lp&noteId=VYOugBMOil`

[7] Kuratowski–Ryll-Nardzewski measurable selection theorem (see [14, Theorem 6.9.3])

where equations marked by (a) and (d) follow from Equation (5), (b) from Lemma 4.1, and (c) from the fact that the inner product space $T_x M$ with $\langle \cdot, \cdot \rangle_x$ is isomorphic to $\mathbb{R}^d$ with the usual dot product. By Theorem 3.8, we have $\widehat{m}(0 \| \tilde{D}^n, K_{\mathbb{R}^d}^{\mathcal{H}}) \to m_{Y|Z}(0)$ in probability. In other words, for all $\epsilon > 0$,

$$\lim_{n \to \infty} \Pr\{|\widehat{m}(0\|\tilde{D}^n, K_{\mathbb{R}^d}^{\mathcal{H}}) - m_{Y|Z}(0)| > \epsilon\} = 0.$$

By Proposition 4.3, we have $m_{Y|Z}(0) = m_{Y|Z}(\exp_x(0)) = m_{Y|Z}(x)$. Therefore,

$$\left\{|\widehat{m}(0\|\tilde{D}^n, K_{\mathbb{R}^d}^{\mathcal{H}}) - m_{Y|Z}(0)| > \epsilon\right\} = \left\{|\widehat{m}(x\|D^n, K_M^{\mathcal{H}}) - m_{Y|X}(x)| > \epsilon\right\}$$

as events. Thus, $\widehat{m}(x\|D^n, K_M^{\mathcal{H}}) \to m_{Y|X}(x)$ converges in probability, proving Theorem 3.2 part 1. As noted in Devroye et al. [22, §2], part 2 of Theorem 3.2 is an immediate consequence of part 1.

# 5  Application to the $d$-Sphere

The $d$-dimensional round sphere is $\mathbb{S}^d := \{x \in \mathbb{R}^{d+1} : x_1^2 + \cdots + x_{d+1}^2 = 1\}$. Here, a *round* sphere assumes that $\mathbb{S}^d$ has the *arc-length metric*:

$$\texttt{dist}_{\mathbb{S}^d}(x,z) = \angle(x,z) = \cos^{-1}(x^\top z) \in [0, \pi]. \tag{7}$$

Let $\mathcal{S}$ be a set and $\sigma : M \to \mathcal{S}$ be a function. The *partition induced* by $\sigma$ is defined by $\{\sigma^{-1}(s) : s \in \texttt{Range}(\sigma)\}$. For example, when $M = \mathbb{S}^d$ and $W \in \mathbb{R}^{(d+1) \times h}$, then the function $\sigma_W : \mathbb{S}^d \to \{\pm 1\}^h$ defined by $\sigma_W(x) = \texttt{sgn}(W^\top x)$ induces a hyperplane arrangement partition.

Let $\mathbb{N} = \{1, 2, \dots\}$ and $\mathbb{N}_0 = \mathbb{N} \cup \{0\}$ denote the positive and non-negative integers.

**Definition 5.1** (Random hyperplane arrangement partition). Let $d \in \mathbb{N}$ and $M = \mathbb{S}^d$. Let $q < 0$ be a negative number, and let $H$ be a random variable with probability mass function $p_H : \mathbb{N}_0 \to [0, 1]$ such that $p_H(h) > 0$ for all $h$. Define the following weighted random partition $\texttt{ha} := (\Theta, \mathfrak{P}, \alpha)$:

1. The parameter space $\Theta = \bigsqcup_{h=0}^{\infty} \mathbb{R}^{(d+1) \times h}$ is the disjoint union of all $(d+1) \times h$ matrices. Element of $\Theta$ are matrices $\theta = W \in \mathbb{R}^{(d+1) \times h}$ where the number of columns $h \in \{0, 1, 2, \dots\}$ varies. By convention, if $h = 0$, the partition $\mathcal{P}_\theta = \mathcal{P}_W$ is the trivial partition $\{\mathbb{S}^d\}$. If $h > 0$, $\mathcal{P}_W$ is the partition induced by $x \mapsto \texttt{sgn}(W^\top x)$.
2. The probability $\mathfrak{P}$ is constructed by the procedure where we first sample $h \sim p_H(h)$, then sample the entries of $W \in \mathbb{R}^{d \times h}$ i.i.d according to $\texttt{Gaussian}(0, 1)$.
3. For $\theta \in \Theta$, define $\alpha(\theta) := \pi^q p_H(h)^{-1}(-1)^h \binom{q}{h}$, where $\binom{q}{h} := \frac{1}{h!} \prod_{j=0}^{h-1} (q - j)$.

Note that $(-1)^h \binom{q}{h} = \frac{1}{h!} \prod_{j=0}^{h-1} (-q + j) > 0$ when $q < 0$.

**Theorem 5.2.** *Let $\texttt{ha} = (\Theta, \mathfrak{P}, \alpha)$ be as in Definition 5.1. Then*

$$K_{\mathbb{S}^d}^{\texttt{ha}}(x,z) = \begin{cases} \angle(x,z)^q & : \angle(x,z) \neq 0 \\ +\infty & : \textit{otherwise.} \end{cases}$$

*When $q = -d$, we have $K_{\mathbb{S}^d}^{\texttt{ha}} = K_{\mathbb{S}^d}^{\mathcal{H}}$ where the right hand side is the manifold-Hilbert kernel.*

*Proof of Theorem 5.2.* Before proceeding, we have the following useful lemma:

**Lemma 5.3.** *Let $\texttt{rp} = (\Theta, \mathfrak{P}, \alpha)$ be a WRP. Let $H$ be a random variable. Let $\theta \sim \mathfrak{P}$. Suppose that for all $x, z \in M$, the random variables $\alpha(\theta)$ and $\mathbb{I}\{x \in \mathcal{P}_\theta[z]\}$ are conditionally independent given $H$. Then we have $K_M^{\texttt{rp}}(x,z) = \mathbb{E}_H\left[\overline{\alpha}(H) \cdot \mathbb{E}_{\theta \sim \mathfrak{P}}[\mathbb{I}\{x \in \mathcal{P}_\theta[z]\}|H]\right]$ where $\overline{\alpha}(h) := \mathbb{E}_{\theta \in \mathfrak{P}}[\alpha(\theta)|H = h]$ for a realization $h$ of $H$.*

The lemma follows immediately from the Definition of $K_M^{\texttt{rp}}(x,z)$ in Equation 3 and the conditional independence assumption. Now, we proceed with the proof of Theorem 5.2.

Let $\phi := \angle(x,z)/\pi$. Let $H \sim p_H$ and $\theta \sim \mathfrak{P}$ be the random variables in Definition 5.1. Note that by construction, the following condition is satisfied: for all $x, z \in M$, the random variables $\alpha(\theta)$

and $\mathbb{I}\{x \in \mathcal{P}_\theta[z]\}$ are conditionally independent given $H$. In fact, $\alpha(\theta) = \pi^q p_H(h)^{-1}(-1)^h \binom{q}{h}$ is constant given $H = h$. Hence, applying Lemma 5.3, we have

$$K^{\mathtt{ha}}_{\mathbb{S}^d}(x, z) = \mathbb{E}_H\left[\overline{\alpha}(H) \cdot \mathbb{E}_{\theta \sim \mathfrak{P}}[\mathbb{I}\{x \in \mathcal{P}_\theta[z]\}|H]\right]$$

$$= \sum_{h=0}^{\infty} \pi^q (-1)^h \binom{q}{h} \cdot \mathbb{E}_{\theta \sim \mathfrak{P}}[\mathbb{I}\{x \in \mathcal{P}_\theta[z]\}|H = h] = \sum_{h=0}^{\infty} \pi^q (-1)^h \binom{q}{h} \cdot \Pr\{x \in \mathcal{P}_\theta[z]|H = h\}.$$

Next, we claim that $\Pr\{x \in \mathcal{P}_\theta[z]|H = h\} = (1 - \phi)^h$. When $h = 0$, $x \in \mathcal{P}_\theta[z]$ is always true since $\mathcal{P}_\theta = \{\mathbb{S}^d\}$ is the trivial partition. In this case, we have $\Pr\{x \in \mathcal{P}_\theta[z]|H = h\} = 1 = (1 - \phi)^0$. When $h > 0$, we recall an identity involving the cosine angle:

**Lemma 5.4** (Charikar [19])**.** *Let $x, z \in \mathbb{S}^d$. Let $w \in \mathbb{R}^{d+1}$ be a random vector whose entries are sampled i.i.d according to* $\mathtt{Gaussian}(0, 1)$. *Then* $\Pr\{\mathtt{sgn}(w^\top x) = \mathtt{sgn}(w^\top z)\} = 1 - (\angle(x, z)/\pi)$.

Let $W = [w_1, \ldots, w_h]$ be as in Definition 5.1 where $w_j$ denotes the $j$-th column of $W$. Then by construction, $w_j$ is distributed identically as $w$ in Lemma 5.4. Furthermore, $w_j$ and $w_{j'}$ are independent for $j, j' \in [h]$ where $j \neq j'$. Thus, the claim follows from

$$\Pr\{x \in \mathcal{P}_\theta[z]|H = h\} \overset{(a)}{=} \Pr\{\mathtt{sgn}(W^\top x) = \mathtt{sgn}(W^\top z)|H = h\}$$

$$\overset{(b)}{=} \prod_{j=1}^{h} \Pr\{\mathtt{sgn}(w_j^\top x) = \mathtt{sgn}(w_j^\top z)\} \overset{(c)}{=} \prod_{j=1}^{h} (1 - \phi) = (1 - \phi)^h.$$

where equality (a) follows from Definition 5.1, (b) from $W \in \mathbb{R}^{(d+1) \times h}$ having i.i.d standard Gaussian entries given $H = h$, and (c) from Lemma 5.4. Putting it all together, we have

$$K^{\mathtt{part}}_{\mathfrak{P},\alpha}(x, z) = \sum_{h=0}^{\infty} \pi^q (-1)^h \binom{q}{h}(1 - \phi)^h = \pi^q \sum_{h=0}^{\infty} \binom{q}{h}(\phi - 1)^h = \angle(x, z)^q.$$

For the last step, we used the fact that for all $q \in \mathbb{R}$ the binomial series $(1 + t)^q = \sum_{h=0}^{\infty} \binom{q}{h} t^h$ converges absolutely for $|t| < 1$ (when $\phi \in (0, 1]$) and diverges to $+\infty$ for $t = -1$ (when $\phi = 0$). $\quad\square$

**Corollary 5.5.** *Let $q := -d$ and $K^{\mathtt{ha}}_{\mathbb{S}^d}$ be as in Theorem 5.2. The infinite-ensemble classifier* $\mathtt{sgn}(\widehat{u}(\bullet \| D^n, K^{\mathtt{ha}}_{\mathbb{S}^d}))$ *(see Equation 4 for definition) has the interpolating-consistent property.*

*Proof.* As observed in Devroye et al. [22, Section 7], for an arbitrary kernel $K$, the $L_1$-consistency of $\widehat{m}(\bullet \| D^n, K)$ for regression implies the consistency for classification of $\mathtt{sgn}(\widehat{u}(\bullet \| D^n, K))$. Furthermore, $\widehat{m}(\bullet \| D^n, K)$ is interpolating for regression implies that $\mathtt{sgn}(\widehat{u}(\bullet \| D^n, K))$ is interpolating for classification. While the argument there is presented in the $\mathbb{R}^d$ case, the argument holds in the more general manifold case *mutatis mutandis*.

Thus, by Theorem 3.2, we have $\mathtt{sgn}(\widehat{u}(\bullet \| D^n, K^{\mathcal{H}}_{\mathbb{S}^d}))$ is consistent for classification, i.e., Equation (1) holds. It is also interpolating since $\widehat{m}(\bullet \| D^n, K)$ is interpolating. By Proposition 5.2, we have $K^{\mathtt{ha}}_{\mathbb{S}^d} = K^{\mathcal{H}}_{\mathbb{S}^d}$. Thus $\mathtt{sgn}(\widehat{u}(\bullet \| D^n, K^{\mathtt{ha}}_{\mathbb{S}^d}))$ is an ensemble method having the interpolating-consistent property. $\quad\square$

## 6 Discussion

We have shown that using the manifold-Hilbert kernel in kernel smoothing regression, also known as Nadaraya-Watson regression, results in a consistent estimator that interpolates the training data on a Riemannian manifold $M$. We proposed *weighted random partition kernels*, a generalization of the unweighted analogous definition by Davies and Ghahramani [21] which provided a framework for analyzing ensemble methods such as random forest via kernels. When $M = \mathbb{S}^d$ is the sphere, we showed that the manifold-Hilbert kernel is a weighted random partition kernel, where the random partitions are induced by random hyperplane arrangements. This demonstrates an ensemble method that has the interpolating-consistent property.

One limitation of this work is the lack of rate of convergence of the ensemble methods. The analogous result for the Nadaraya-Watson regression have been obtained by Belkin et al. [8]. However, it is not

clear if the kernels used in [8] are weighted random partition kernels. Resolving this is an interesting future direction.

Similar to PERT [20], another limitation of this work is that the base classifiers in the ensemble are data-independent. Such ensemble methods in these line of work (including ours) are easier to analyze than the data-*dependent* ensemble methods used in practice. See Biau and Scornet [12] and [13] for an in-depth discussion. We believe our work offers one theoretical basis towards understanding generalization in the interpolation regime of ensembles of histogram classifiers over data-dependent partitions, e.g., decision trees à la CART [17].

## Acknowledgements

The authors were supported in part by the National Science Foundation under awards 1838179 and 2008074. The authors would like to thank Vidya Muthukumar for helpful discussions, as well as an anonymous reviewer for bringing to our attention facts from geometric measure theory that provided context for the ad-hoc technical result Proposition 4.2.

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
