# Supplemental Materials: Consistent Interpolating Ensembles via the Manifold-Hilbert Kernel

## A.1 Basics of Riemannian Manifolds

In this section, we review the main concepts from Riemannian manifold theory essential to this work. Our main references are Sakai [36] and Do Carmo [23]. Throughout, $d \in \mathbb{N}$ denotes the dimension. We use the word *smooth* to mean infinitely differentiable.

**Manifolds.** A smooth *manifold* $M$ of dimension $d$ is a Hausdorff, second countable topological space together with an *atlas*: a set $\texttt{Atlas} := \{(U_\alpha, \varphi_\alpha)\}_{\alpha \in A}$ where 1). $\{U_\alpha\}_{\alpha \in A}$ is an open cover of $M$, 2). for each $\alpha \in A$, $\varphi_\alpha : U_\alpha \to \varphi_\alpha(U_\alpha) \subseteq \mathbb{R}^d$ is a homeomorphism onto its image, and 3). $\varphi_\alpha \circ \varphi_\beta^{-1} : \varphi_\beta(U_\alpha \cap U_\beta) \to \varphi_\alpha^{-1}(U_\alpha \cap U_\beta)$ is smooth for each pair $\alpha, \beta \in A$. An element $(U, \varphi)$ of $\texttt{Atlas}$ is called a *chart*.

**Smooth maps.** A real-valued function $f : M \to \mathbb{R}$ is a *smooth function* if $f \circ \varphi^{-1}$ is smooth (in the elementary calculus sense) for all charts $(U, \varphi)$. The set of all smooth functions is denoted $\texttt{Fn}(M)$, which forms an $\mathbb{R}$-vectorspace. Let $N$ be another smooth manifold with atlas $\mathcal{B}$. A function $\Phi : M \to N$ is a *smooth map* if $g \circ \Phi \in \texttt{Fn}(M)$ for all $g \in \texttt{Fn}(N)$.

**Tangent space.** Let $x \in M$. A *derivation at $x$* is a linear function $v : \texttt{Fn}(M) \to \mathbb{R}$ satisfying the *product rule*: $v[fg] = f(x)v[f] + g(x)v[g]$ for all $f, g \in \texttt{Fn}(M)$. The *tangent space at $x$*, denoted $T_x M$, is the vector space of all derivations at $x$. Elements of $T_x M$ are referred to as *tangent vectors* at $x$. For a given chart $(U, \varphi)$ where $x \in U$, define a derivation at $x$, denoted $\partial_i|_x$, by $f \mapsto \frac{d(f \circ \varphi^{-1})}{dz_i}(\varphi(x))$ where $\frac{d}{dz_i}$ is the $i$-th partial derivative in ordinary calculus. It is a fact that $\{\partial_i|_x : i = 1, \ldots, d\}$ is a basis for $T_x M$.

Although the above definition of a tangent vector is abstract, it can be concretely interpreted in terms of derivative along a curve. Let $a < t_0 < b$ be real numbers. A *curve through $x$* is a smooth map $\gamma : (a, b) \to M$ such that $\gamma(t_0) = x$. Then $\texttt{Fn}(M) \ni f \mapsto \frac{d}{dt} f(\gamma(t))|_{t=t_0} \in \mathbb{R}$ defines a derivation at $x$. Oftentimes, this derivation is denoted $\dot{\gamma}(t_0) \in T_x M$

**Riemannian metric.** The *tangent bundle* is the set $TM := \bigcup_x T_x M$, which itself is a smooth manifold of dimension $2d$. A *vector field on $M$* is a smooth map $\mathsf{V} : M \to TM$ such that $\mathsf{V}(x) \in T_x M$ for all $x \in M$. The set of all vectors fields on $M$ is denoted $\texttt{Vf}(M)$.

A *Riemannian metric* on $M$ is a choice of an inner product $\langle \cdot, \cdot \rangle_x$ (and thus, a norm $\| \cdot \|_x$) on $T_x M$ for each $x \in M$ such that the function $M \to \mathbb{R}$ given by $x \mapsto \langle \mathsf{V}(x), \mathsf{U}(x) \rangle_x$ is smooth for all $\mathsf{V}, \mathsf{U} \in \texttt{Vf}(M)$. As shorthands, when $x$ is clear from context, we drop the subscripts and simply write $\langle \cdot, \cdot \rangle$ and $\| \cdot \|$ instead. Choosing an orthonormal basis for $T_x M$ with respect to $\langle \cdot, \cdot \rangle_x$ for each $x$, we can identify $T_x M$ with $\mathbb{R}^d$ with the ordinary dot inner product.

Let $x \in M$ and $(U, \varphi)$ be a chart such that $x \in U$. Define $g_{ij}(x) = \langle \partial_i|_x, \partial_j|_x \rangle_x$. Denote by $G(x)$ the $d \times d$ positive definite matrix $[g_{ij}(x)]_{ij}$. Below, we will refer to the function $G : U \to \mathbb{R}^{d \times d}$ as the *coordinate representation* of the Riemannian metric. Define $g^{ij}(x) := [G(x)^{-1}]_{ij}$. The *Christoffel symbols* with respect to $(U, \varphi)$ are defined by $\Gamma_{ij}^k := \frac{1}{2} \sum_{\ell=1}^d g^{k\ell} (\partial_i|_x g_{j\ell} + \partial_j|_x g_{i\ell} - \partial_\ell|_x g_{ij})$. Note that $g_{k\ell}$, $g^{k\ell}$, $G$, $\Gamma_{ij}^k$, and $\partial_i|_x g_{j\ell}$ are all functions with domain $U$.

**Geodesics.** Fix a chart $(U, \varphi)$. Consider a smooth curve $\gamma : [a, b] \to U$. Let $\zeta_i(t) := [\varphi(\gamma(t))]_i$ be the $i$-th component functions. The curve $\gamma$ is a *geodesic* if $\zeta$ is a solution to the following system of second order ordinary differential equations (ODEs): $\frac{d^2 \zeta_i}{dt^2} + \sum_{j,\ell=1}^d \Gamma_{j\ell}^i \circ \gamma \frac{d\zeta_j}{dt} \frac{d\zeta_\ell}{dt} = 0$ for all $i = 1, \ldots, d$ at all time $t \in [a, b]$.

Geodesics are minimizers of the so-called *energy functional* $E(\gamma) = \frac{1}{2} \int_a^b \| \dot{\gamma}(t) \|_{\gamma(t)}^2 dt$. The above system of ODEs are the analog of the "first derivative test" for local minimizers of $E$. Thus, geodesics are defined independently of the choice of the chart.

**Exponential map.** For $x \in M$ and $v \in T_x M$, there exists $\epsilon > 0$ and a unique geodesic curve $\gamma_v : [-\epsilon, \epsilon] \to M$ such that $\gamma_v(0) = x$ and $\dot{\gamma}_v(0) = v$. This follows from the existence and uniqueness of the solution to an ODE given initial conditions where the ODE is as discussed above.

Note that although geodesics are previously defined in $U$ where $(U, \varphi)$ is a chart, they can be extended outside of $U$ using additional charts.

Let $x \in M$ and $v \in T_x M$ be fixed and let $\gamma_v : [-\epsilon, \epsilon] \to M$ be as in the preceding paragraph. If $\|v\|_x \leq \epsilon$, then define $\exp_x(v) := \gamma_v(1)$. A fundamental fact is that $\exp_x$, known as the *exponential map at $x$*, can be defined on an open set of $T_x M$ containing the origin.

**Distance function.** Let $x, \xi \in M$ and $a < b$ be real numbers. A *piecewise smooth curve* from $x$ to $\xi$ is a piecewise smooth map $\gamma : [a, b] \to M$ such that $\gamma(a) = x$ and $\gamma(b) = \xi$. Assume that $M$ is connected. Then for all $x, \xi \in M$, there exists a piecewise smooth curve from $x$ to $\xi$. The *length of $\gamma$* is defined as $\mathtt{len}(\gamma) := \int_a^b \|\dot\gamma(t)\|_{\gamma(t)} dt$. Define $\mathtt{dist}_M(x, \xi) := \inf\{\mathtt{len}(\gamma) : \gamma$ is a piecewise smooth curve from $x$ to $\xi\}$, which is a metric on $M$ in the sense of metric spaces (see [36, Proposition 1.1]). For $x \in M$ and $r \in (0, \infty)$, the open ball centered at $x$ of radius $r$ is denoted $\mathtt{B}_x(r, M) := \{z \in M : \mathtt{dist}_M(x, z) < r\}$.

**Complete Riemannian manifolds.** A Riemannian manifold is *complete* if it is a complete metric space under the metric $\mathtt{dist}_M$. The Hopf-Rinow theorem ([23, Ch. 8, Theorem 2.8]) states that if $M$ is connected and complete, then the exponential $\exp_x$ can be defined on the entire $T_x M$.

## A.2 Proof of Lemma 4.1

This section uses definitions and notations introduced in Section 4.1. In particular, recall the cut locus $C_x$, the tangent cut locus $\tilde{C}_x$, the interior set $I_x$ and the tangent interior set $\tilde{I}_x$. The proof of Lemma 4.1 is presented towards the end of the section. At this point, we compile some facts from various sources about the cut locus.

**Lemma A.1.** *For all $x \in M$, we have*

1. *$C_x$ is a closed subset of $M$ (Hebda [29, Proposition 1.2]).*
2. *$I_x \cap C_x = \emptyset$ and $I_x \cup C_x = M$ (Sakai [36, Ch II, Lemma 4.4 (1)])*
3. *$I_x$ is an open subset of $M$ (immediate from 1 and 2 above)*
4. *$\exp_x : \tilde{I}_x \to I_x$ is a diffeomorphism ([36, Ch II, Lemma 4.4 (2)])*
5. *$\lambda_M(C_x) = 0$, where $\lambda_M$ is the Riemann-Lebesgue measure ([36, Lemma 4.4 (3)])*
6. *$\tau_x$ is continuous and $\inf_{u \in U_x M} \tau_x(u) > 0$ ([36, Ch II, Propositions 4.1 (2) and 4.13 (1)])*

While the following lemma is elementary, we provide a proof since we could not find one in the literature.

**Lemma A.2.** *For all $x \in M$, the (topological) closure of $\tilde{I}_x$ in $T_x M$ is $\tilde{D}_x$. Furthermore, for all $x \in M$, we have $\exp_x(\tilde{D}_x) = M$.*

*Proof of Lemma A.2.* Take a convergent sequence $\{t_i u_i\}_{i \in \mathbb{N}} \subseteq \tilde{I}_x$ where $u_i \in U_x M$ and $0 \leq t_i < \tau_x(u_i)$. Let $v^* = \lim_i t_i u_i$. Our goal is to show that $v^* \in \tilde{D}_x = \tilde{I}_x \cup \tilde{C}_x$.

Since $U_x M$ is compact, we may assume that $u^* := \lim_i u_i$ exists after passing to a subsequence if necessary. Furthermore, $\|t_i u_i\|_x = t_i$ implies that $t^* := \lim_i t_i$ exists as well (i.e., $t^* < \infty$). Hence, $v^* = t^* u^*$.

Consider the case that $\tau_x(u^*) = \infty$. Then $0 \leq t^* < \tau_x(u^*)$ implies that $v^* = t^* u^* \in \tilde{I}_x$. For the other case that $t(u) < \infty$, we first note that $t_i u_i \in \tilde{I}_x$ implies that $t_i < \tau_x(u_i)$. Taking the limit of both sides, we have $t^* = \lim_i t_i \leq \lim_i \tau_x(u_i) = \tau_x(u^*)$. Note that the last limit can be exchanged since $\tau_x$ is continuous (Lemma A.1 part 6). Thus, either $t^* < \tau_x(u^*)$ in which case $v^* \in \tilde{I}_x$, or $t^* = \tau_x(u^*)$ in which case $v^* = \tau_x(u^*) u^* \in \tilde{C}_x$.

For the "furthermore" part, note that

$$\exp_x(\tilde{D}_x) = \exp_x(\tilde{I}_x \cup \tilde{C}_x) = \exp_x(\tilde{I}_x) \cup \exp_x(\tilde{C}_x) = I_x \cup C_x = M$$

where the last equality is Lemma A.1 part 2. $\square$

*Proof of Lemma 4.1.* Denote by $\mathrm{cl}(T_x M)$ the set of closed subsets of $T_x M$. Define $\psi : M \to \mathrm{cl}(T_x M)$ by $\psi(\xi) := \{x \in \tilde{D}_x : \exp_x(x) = \xi\} = \exp_x^{-1}(\xi) \cap \tilde{D}_x$. Note that $\psi(\xi)$ is a closed set by Lemma A.2.

We claim that $\psi$ is *weakly-measurable*, i.e., for every open set $\tilde{U} \subseteq T_x M$, the subset of $M$ defined by $\{\xi \in M : \psi(\xi) \cap \tilde{U} \neq \emptyset\}$ is Borel. To see this, note that

$$
\begin{aligned}
&\{\xi \in M : \psi(\xi) \cap \tilde{U} \neq \emptyset\} \\
&= \{\xi \in M : \exp_x^{-1}(\xi) \cap \tilde{D}_x \cap \tilde{U} \neq \emptyset\} \\
&= \{\xi \in M : \exp_x(\tilde{D}_x \cap \tilde{U}) \ni \xi\} \\
&= \exp_x(\tilde{D}_x \cap \tilde{U}).
\end{aligned}
$$

As inner product spaces, $T_x M$ and $\mathbb{R}^d$ are isomorphic (see Section A.1-Riemannian metric). Since, $T_x M$ and $\mathbb{R}^d$ are homeomorphic as topological spaces, $\mathbb{R}^d$ being locally compact implies $T_x M$ is locally compact as well. Thus, we can write $\tilde{U} = \bigcup_{i \in \mathbb{N}} \tilde{K}_i$ as a countable union of compact sets $\tilde{K}_i \subseteq T_x M$. Furthermore, $\tilde{D}_x \cap \tilde{U} = \bigcup_{i \in \mathbb{N}} \tilde{D}_x \cap \tilde{K}_i$ and so $\exp_x(\tilde{D}_x \cap \tilde{U}) = \bigcup_{i \in \mathbb{N}} \exp_x(\tilde{D}_x \cap \tilde{K}_i)$.

Since $\exp_x$ is continuous, $\exp_x(\tilde{D}_x \cap \tilde{K}_i)$ is a compact subset of $M$, and hence closed and bounded by the Hopf-Rinow theorem ([23, Ch. 8, Theorem 2.8]). Thus, $\exp_x(\tilde{D}_x \cap \tilde{U}) = \bigcup_{i \in \mathbb{N}} \exp_x(\tilde{D}_x \cap \tilde{K}_i)$ is a countable union of closed sets, which is Borel. This proves the claim that $\psi$ is weakly Borel measurable.

By the Kuratowski–Ryll-Nardzewski measurable selection theorem (see [14, Theorem 6.9.3]), there exists a Borel measurable function $M \to T_x M$, which we denote by $\log_x$, such that $\log_x(\xi) \in \psi(\xi) = \exp_x^{-1}(\xi)$ for all $\xi \in M$, as desired. By construction, $\log_x(\xi) \in \exp_x^{-1}(\xi)$ for all $\xi \in M$, and so $\exp_x(\log_x(\xi)) = \xi$ is immediate.

For the "furthermore" part, let $\xi \in M$ be arbitrary and let $z := \log_x(\xi) \in \tilde{D}_x$. Let $\{z_i\} \subseteq \tilde{I}_x$ be a sequence such that $\lim_i z_i = z$. By Equation (6), we have $\text{dist}_M(x, \exp_x(z_i)) = \|z_i\|_x$. By continuity of $\text{dist}_M$ and $\exp_x$, we have $\text{dist}_M(x, \xi) = \text{dist}_M(x, \exp_x(z)) = \lim_i \text{dist}_M(x, \exp_x(z_i))$. To conclude, we have $\lim_i \text{dist}_M(x, \exp_x(z_i)) = \lim_i \|z_i\|_x = \|z\|_x = \|\log_x(\xi)\|_x$, as desired. $\square$

## A.3 Proof of Proposition 4.2

Recall from Section A.1-Riemannian metric, given a chart $(U, \varphi)$, one can define the matrix-valued function $G : U \to \mathbb{R}^{d \times d}$ referred to earlier as the coordinate representation of the Riemannian metric. Now, Lemma A.1 part 3 states that $I_x$ is an open neighborhood of $x$. Furthermore, $\tilde{I}_x$ is an open subset of $T_x M$, which is identified with $\mathbb{R}^d$ using an orthonormal basis (see Section A.1-Riemannian metric). Hence, $\{(I_x, \log_x|_{I_x})\}_{x \in M}$ is an atlas of $M$ (see Section A.1-Manifolds).

**Definition A.3.** The chart $(I_x, \log_x|_{I_x})$ is called a *normal coordinate system* at $x$. Let $G : I_x \to \mathbb{R}^{d \times d}$ be the coordinate representation of the Riemannian metric for this chart. To emphasize the dependency on $x$, we write $G_x := G$. Denote by $G_x^\perp : M \to \mathbb{R}^{d \times d}$ the zero extension of $G_x$ to the rest of $M$, i.e., $G_x^\perp(\xi) = G_x(\xi)$ for $\xi \in I_x$ and $G_x^\perp(\xi)$ is the zero matrix for $\xi \notin I_x$.

The normal coordinate system has the property that $G_x(x) = G_x^\perp(x)$ is the identity matrix. This is the result of Sakai [36, Ch. II §2 Exercise 4].

**Lemma A.4** (Change-of-Variables). *Let $x \in M$ be fixed. Define the function $\nu_x : M \to \mathbb{R}$ by $\nu_x(\xi) = \sqrt{|\det G_x^\perp(\xi)|}$ where $G_x^\perp$ is as in Definition A.3. Then $\nu_x$ is Borel-measurable. Furthermore, $\nu_x$ satisfies the following property: Let $f : M \to \mathbb{R}$ be an absolutely integrable function. Define the function*

$$h : T_x M \to \mathbb{R} \quad by \quad h(z) := f(\exp_x(z)) \cdot \nu_x(\exp_x(z)).$$

*Then* (i) $h(0) = f(x)$ *and* (ii) *for all Borel set $\tilde{B} \subseteq T_x M$ we have $\int_B f d\lambda_M = \int_{\tilde{B}} h d\lambda$ where $B := \exp_x(\tilde{B} \cap \tilde{I}_x)$.*

*Proof of Lemma A.4.* We first show that $\nu_x$ is Borel-measurable. Recall that $G_x^\perp : M \to \mathbb{R}^{d \times d}$ is the zero extension of $G_x : I_x \to \mathbb{R}$, which is by definition smooth (see Section A.1-Riemannian metric). In particular, $G_x : I_x \to \mathbb{R}$ is continuous and so $\sqrt{\det(G_x(\bullet))}$ is Borel-measurable. Now, note that $\sqrt{\det(G_x^\perp(\bullet))}$ is the zero extension of $\sqrt{\det(G_x(\bullet))}$ from $I_x$ to $M$. Hence, $\sqrt{\det(G_x^\perp(\bullet))}$, which is $\nu_x$ by definition, is Borel-measurable.

Next, we prove the "Furthermore" part (i). Note that $\exp_x(0) = x$. Moreover, $G_x^\perp(x) = G_x(x)$ is the identity matrix as asserted after Definition A.3 (see Sakai [36, Ch. II §2 Exercise 4]). Thus, $h(0) = f(\exp_x(0))\sqrt{|\det G_x^\perp(\exp_x(0))|} = f(x)\sqrt{1} = f(x)$, as desired.

For the "Furthermore" part (ii), we first note that $\tilde{B} = (\tilde{B} \cap \tilde{I}_x) \cup (\tilde{B} \cap \tilde{C}_x)$ expresses $\tilde{B}$ as a disjoint union. Thus, $B = \exp_x(\tilde{B}) = \exp_x(\tilde{B} \cap \tilde{I}_x) \cup \exp(\tilde{B} \cap \tilde{C}_x)$ expresses $B$ as a disjoint union as well. Moreover, $\exp(\tilde{B} \cap \tilde{C}_x) \subseteq \exp(\tilde{C}_x) = C_x$, which has $\lambda_M$-measure zero (Lemma A.1 part 5).

Recall that $\lambda$ is the shorthand for the ordinary Lebesgue measure $\lambda_{\mathbb{R}^d}$ (see paragraph right after Definition 3.1). Now, we directly compute to obtain the formula

$$
\begin{aligned}
\int_{\tilde{B}} h\, d\lambda &= \int_{\tilde{B} \cap \tilde{I}_x} f \circ \exp_x \sqrt{|\det(G_x^\perp \circ \exp_x)|}\, d\lambda \\
&= \int_{\log_x(\exp_x(\tilde{B} \cap \tilde{I}_x))} f \circ \exp_x \sqrt{|\det(G_x^\perp \circ \exp_x)|}\, d\lambda \\
&= \int_{\exp_x(\tilde{B} \cap \tilde{I}_x)} f\, d\lambda_M \qquad \because \text{ Amann and Escher [1, Ch XII, Thm 1.10]} \\
&= \int_{\exp_x(\tilde{B} \cap \tilde{I}_x)} f\, d\lambda_M + \int_{\exp_x(\tilde{B} \cap \tilde{C}_x)} f\, d\lambda_M \\
&= \int_B f\, d\lambda_M,
\end{aligned}
$$

as desired. $\qquad\qquad\square$

**Proposition A.5.** *Let $x \in M$ be fixed. Let $X$ be a random variable on $M$ with density $f_X$ where the underlying probability space is $(\Omega, \mathbb{P}, \mathcal{A})$ (see Definition 3.3). Define $Z := \log_x(X)$. Then $Z$ is a random variable on $T_x M$ such that for all events $E \in \mathcal{A}$ and Borel sets $\tilde{B} \subseteq T_x M$ we have $\Pr(E \cap \{Z \in \tilde{B}\}) = \Pr(E \cap \{X \in \exp_x(\tilde{B} \cap \tilde{I}_x)\})$,*

*Proof of Proposition A.5.* To start with, we have

$$
\begin{aligned}
&\Pr(E \cap \{Z \in \tilde{B}\}) \\
&= \Pr(E \cap \{Z \in \tilde{B} \cap \tilde{D}_x\}) \qquad \because \log_x(M) \subseteq \tilde{D}_x \\
&= \Pr(E \cap \{Z \in \tilde{B} \cap \tilde{I}_x\}) + \Pr(E \cap \{Z \in \tilde{B} \cap \tilde{C}_x\}) \qquad \because \tilde{D}_x = \tilde{I}_x \cup \tilde{C}_x, \ \emptyset = \tilde{I}_x \cap \tilde{C}_x \\
&= \Pr(E \cap \{\log_x(X) \in \tilde{B} \cap \tilde{I}_x\}) + \Pr(E \cap \{\log_x(X) \in \tilde{B} \cap \tilde{C}_x\}).
\end{aligned}
$$

Since $\exp_x : \tilde{I}_x \to I_x$ is a diffeomorphism (Lemma A.1-part 4) with inverse $\log_x$, we have

$$
E \cap \{\log_x(X) \in \tilde{B} \cap \tilde{I}_x\} = E \cap \{X \in \exp_x(\tilde{B} \cap \tilde{I}_x)\}
$$

as sets. On the other hand,

$$
E \cap \{\log_x(X) \in \tilde{B} \cap \tilde{C}_x\} \subseteq \{X \in C_x\}.
$$

Finally, $\Pr(X \in C_x) = \int_{C_x} f_X\, d\lambda_M = 0$ since $C_x$ has $\lambda_M$-measure zero (Lemma A.1-part 5). $\quad\square$

*Proof of Proposition 4.2 part (i).* Recall that $\lambda$ is the shorthand for the ordinary Lebesgue measure $\lambda_{\mathbb{R}^d}$ (see paragraph right after Definition 3.1). Let $E = \Omega$ in Proposition A.5. Then we have

$$
\begin{aligned}
&\Pr(Z \in \tilde{B}) \\
&= \Pr(X \in \exp_x(\tilde{B} \cap \tilde{I}_x)) \qquad \because \text{ Part (i)} \\
&= \int_{\exp_x(\tilde{B} \cap \tilde{I}_x)} f_X\, d\lambda_M \qquad \because f_X \text{ is the density of } X \\
&= \int_{\tilde{B} \cap \tilde{I}_x} (f_X \circ \exp_x) \cdot (\nu_x \circ \exp_x)\, d\lambda \qquad \because \text{ Lemma A.4} \\
&= \int_{\tilde{B}} f_Z\, d\lambda \qquad \because \text{ Definition of } f_Z
\end{aligned}
$$

By assumption, $f_X$ is Borel-measurable. By Lemma A.4, $\nu_x$ is Borel-measurable. Since $\exp_x$ is continuous, we have that both $f_X \circ \exp_x$ and $\nu_x \circ \exp_x$ are Borel-measurable. This proves that $\tilde{f}_Z$ is Borel-measurable. Hence, the integrand is Borel-measurable and a density function for $Z$. □

*Proof of Proposition 4.2 part (ii).* Recall that $\lambda$ is the shorthand for the ordinary Lebesgue measure $\lambda_{\mathbb{R}^d}$ (see paragraph right after Definition 3.1). By Lemma A.1 part 6, we have $\tau_x^* := \inf_{u \in U_x M} \tau_x(u) > 0$. Now, let $r \in (0, \tau_x^*)$. By the definition of $r$, we have $\mathtt{B}_x(r, M) \subseteq \tilde{I}_x$. Hence letting $z = \log_x(\xi)$ for $\xi \in \mathtt{B}_x(r, M)$, by Equation (6) we have

$$\mathtt{dist}_M(x, \xi) = \mathtt{dist}_M(x, \exp_x(z)) = \|z\|_x. \tag{8}$$

Thus,

$$\log_x(\mathtt{B}_x(r, M)) = \{z \in T_x M : \|z\|_x < r\} = \mathtt{B}_0(r, T_x M) \tag{9}$$

and

$$\mathtt{B}_x(r, M) = \exp_x(\mathtt{B}_0(r, T_x M)). \tag{10}$$

Thus, by Lemma A.4, we have

$$\int_{\mathtt{B}_x(r,M)} f d\lambda_M = \int_{\mathtt{B}_0(r,T_xM)} h d\lambda. \tag{11}$$

Before proceeding, we need the following lemma:

**Lemma A.6.** *For all $x \in M$, we have $\lim_{r \to 0} \frac{\lambda_M(\mathtt{B}_x(r,M))}{\lambda(\mathtt{B}_0(r,T_xM))} = 1$.*

*Proof of Lemma A.6.* Let $\omega_d := \pi^{d/2}/\Gamma(\frac{d}{2}+1)$ be the volume of the unit ball in $\mathbb{R}^d$ where $\Gamma$ is the gamma function. Then $\lambda(\mathtt{B}_0(r, T_x M)) = \omega_d r^d$. Next, [36, Ch II.5 Exercise 3] states that

$$\lim_{r \to 0} \frac{r^d \omega_d - \lambda_M(\mathtt{B}_x(r, M))}{r^{d+2}} = \frac{\omega_d}{6(d+2)} S_x$$

where $S_x \in \mathbb{R}$ is a constant that depends only on $x$ (it is the scalar curvature of $M$ at $x$). By simple algebra, the above yields

$$0 = \lim_{r \to 0} \frac{1}{r^2} \left( 1 - \frac{\lambda_M(\mathtt{B}_x(r, M))}{\omega_d r^d} - \frac{S_x r^2}{6(d+2)} \right)$$

In particular, we have $\lim_{r \to 0} 1 - \frac{\lambda_M(\mathtt{B}_x(r,M))}{\omega_d r^d} = 0$, as desired. □

Now we continue with the proof of Proof of Proposition 4.2 part (ii). We observe that

$$
\begin{aligned}
f(x) &= \lim_{r \to 0} \frac{\int_{\mathtt{B}_x(r,M)} f d\lambda_M}{\lambda_M(\mathtt{B}_x(r, M))} \quad \because x \text{ is a Lebesgue point of } f \\
&= \lim_{r \to 0} \frac{\int_{\mathtt{B}_0(r,T_x(M))} h d\lambda}{\lambda_M(\mathtt{B}_x(r, M))} \quad \because \text{ definition of } h \text{ and equation (11)} \\
&= \lim_{r \to 0} \frac{\int_{\mathtt{B}_0(r,T_xM)} h d\lambda}{\lambda_M(\mathtt{B}_x(r, M))} \frac{\lambda_M(\mathtt{B}_x(r, M))}{\lambda(\mathtt{B}_0(r, T_xM))} \quad \because \text{ Lemma A.6} \\
&= \lim_{r \to 0} \frac{\int_{\mathtt{B}_0(r,T_xM)} h d\lambda}{\lambda(\mathtt{B}_0(r, T_xM))}.
\end{aligned}
$$

Since $f(x) = h(0)$ (Lemma A.4) , we've shown that

$$g(0) = \lim_{r \to 0} \frac{\int_{\mathtt{B}_0(r,T_xM)} h d\lambda}{\lambda(\mathtt{B}_0(r, T_xM))}.$$

Thus, 0 is a Lebesgue point of $h$, as desired. □

## A.4 Proof of Proposition 4.3

Recall that $\lambda$ is the shorthand for the ordinary Lebesgue measure $\lambda_{\mathbb{R}^d}$ (see paragraph right after Definition 3.1). Let $A \subseteq \mathbb{R}$ and $\tilde{B} \subseteq T_x M$ be Borel subsets. Then

$$\int_{\tilde{B}} P_{Y|Z}(A|z) f_Z(z) d\lambda(z)$$

$$= \int_{\tilde{B}} P_{Y|X}(A|\exp_x(z)) f_Z(z) d\lambda(z) \qquad \because \text{Definition of } P_{Y|Z=z}$$

$$= \int_{\exp_x(\tilde{B} \cap \tilde{I}_p)} P_{Y|X}(A|x) f_X(x) d\lambda_M(x) \qquad \because \text{Lemma A.4 and Proposition 4.2 (ii)}$$

$$= \Pr(Y \in A, X \in \exp_x(\tilde{B} \cap \tilde{I}_x))$$

$$= \Pr(Y \in A, Z \in \tilde{B}) \qquad \because \text{Proposition 4.2 (i) with } E := \{Y \in A\}$$

This proves that $P_{Y|Z}(\cdot|\cdot)$ is a conditional probability for $Y$ given $Z$.