# OpenReview forum: "Consistent Interpolating Ensembles via the Manifold-Hilbert Kernel"
_NeurIPS.cc/2022/Conference — NeurIPS 2022 Accept_

### Official Review · Reviewer_SNA2 · 2022-06-28

**Rating:** 6
**Confidence:** 1
**Soundness:** 3 good
**Presentation:** 2 fair
**Contribution:** 3 good

**Summary:**

The authors derive an ensemble classification method for interpolating manifold-valued training data that they denote manifold-Hilbert kernel. The kernel is based on the Riemannian distances. The authors prove a consistency result with the kernel and derive a specific realization of the kernel on spheres.

**Questions:**

no questions

**Limitations:**

yes

**Strengths And Weaknesses:**

Strengths:
* the authors derive an ensemble method that has the consistent-interpolating property
* the authors define the manifold-Hilbert kernel
* the authors derive a specific realization of the kernels on spheres

Weaknesses:
* the paper is very technical and very hard to read.

As a non-specialist in this area, I was not able to confirm correctness of the presented results However, I have no reason to believe that the paper is not technically correct.

I have a hard time evaluating the novelty. Proving consistency in this setting seems like a good result, but I don't know the literature well enough to judge how significant this is.

There is no experimental evaluation. Since the paper is purely theoretical, I believe this is OK.

---

> ### Author Response · Authors · 2022-08-01
> **Readability and novelty of our contributions**
>
> We thank the reviewer for the positive comments and the constructive criticisms.
>
>
> > the paper is very technical and very hard to read.
>
>
> We added an additional figure (Figure 1 in the new revised version) to Section 1.1 and 1.2 to improve the exposition.
> Hopefully this addresses some of the readability issues.
>
>
> > I have a hard time evaluating the novelty. Proving consistency in this setting seems like a good result, but I don't know
> the literature well enough to judge how significant this is.
>
> Our work links together several ideas that appeared in the literature in different context in a non-trivial way.
> For instance, while the Riemannian logarithm has been studied by the manifold optimization community,
> we found that it is a key tool for proving consistency for learning theory on a manifold.
> Random partitions kernels (RPKs) have also been studied in works analyzing ensembles.
> Our innovation is in realizing that the weighted variant of the RPK
> can be used to establish consistency for interpolating algorithms.
>
> Our contributions to Riemannian geometry are straightforward extension of well-known results on Euclidean spaces (remark added on Line 252).
> However, our main innovation is putting them together in a nontrivial way to reduce our main result to that of
> Devroye, Gyorfi, Krzyzak (1998).
> These supporting lemmas, while simple, have not appeared in the literature in a way that immediately applies in
> our setting.
> We prove them rigorously for the sake of completeness.
> Our main result also opens the door for future work to simply design kernels on the manifold.
> As such, we believe it is a valuable contribution to rigorously prove these supporting lemmas.
>
> ---
>
> ### References
>
>
> Devroye, Gyorfi, Krzyzak (1998) The Hilbert kernel regression estimate, Journal of Multivariate Analysis.

---

### Official Review · Reviewer_w6Vf · 2022-07-06

**Rating:** 7
**Confidence:** 1
**Soundness:** 4 excellent
**Presentation:** 4 excellent
**Contribution:** 3 good

**Summary:**

In this paper, the authors first define the manifold-Hilbert kernel on a complete, connected, smooth Riemannian manifold of dimension d, as the d-th power reciprocal of the manifold distance between two points. The main result of the paper (Theorem 3.2) is that the Nadaraya-Watson estimator of the conditional mean of Y given X based on this manifold-Hilbert kernel, which by design interpolates the data, is consistent, both pointwise at almost every point and in terms of the L1 distance with respect to the density of X, assumed to exist.

In Section 5, the authors move onto a particular case of the manifold being the d-dimensional sphere, and it is shown in Theorem 5.2 that the random hyperplane arrangement partition, which corresponds to an ensemble method, actually coincides with the manifold-Hilbert kernel proposed in earlier parts of the paper, thereby showing that ensemble classifiers of this form are interpolating and consistent.

**Questions:**

I am not familiar with ensemble methods, having never directly worked with them or used them. Is the form of ensemble used in this work, namely the one based on weighted random partition in (4), a general one in which many ensemble methods fall under? Or is this a particular form, which could limit the scope of your contribution?

**Limitations:**

The contributions of this paper are theoretical in nature, thence I do not see any reason to discuss potential negative societal impact.

Minor comments:

line 77: "When show that" -> "We show that"
line 135: $\epsilon>0$ is a real positive number, not an open set, I presume?


**Strengths And Weaknesses:**

I do not actively work with differential/Riemannian geometry in my research, and beyond basic notions which I acquired a number of years ago, I am not familiar with the concepts used in this paper, and due to time constraints, I could not read up on all the background material required to go through every step of the proofs and certify their correctness.

Strengths

I believe the paper is tackling an important problem, although as someone who does not work directly in relevant areas, my judgment on the significance of the results is not well-grounded.

The paper is technical, but in spite of this, the clarity of the mathematical presentation makes it a pleasure to read. Despite my lack of required background to fully judge its merit, this helped me form a more positive picture of the work, and in parts where I did try to go through the mathematics, helped convince me of its soundness.

Weaknesses

The main weakness that I can see is the lack of convergence rates. Only consistency is shown in the infinite-sample limit, but I think results would be much stronger if uniform rates were given over some class of distributions.

---

> ### Author Response · Authors · 2022-08-01
> **Rate of convergence and scope of contribution**
>
> We thank the reviewer for the positive comments and the constructive criticisms.
>
>
> > The main weakness that I can see is the lack of convergence rates. Only consistency is shown in the infinite-sample limit,
> but I think results would be much stronger if uniform rates were given over some class of distributions.
>
> We agree with the reviewer that this is an interesting direction.
> We have explored the kernels studied in Belkin, Rakhlin, Tsybakov (2019) which have the properties as the reviewer suggested.
>
> However, we were not able to show whether or not such kernels as in Belkin, Rakhlin, Tsybakov (2019) are (weighted) random partition kernels.
> Resolving this open question is a major challenge, which we added to the Discussions section beginning on Line 333.
>
> > I am not familiar with ensemble methods, having never directly worked with them or used them. Is the form of
> ensemble used in this work, namely the one based on weighted random partition in (4), a general one in which many
> ensemble methods fall under? Or is this a particular form, which could limit the scope of your contribution?
>
> Ensemble of random partitions indeed encompass many of the ensemble methods of trees, i.e., random forest, used in practice.
> For instance, see Davies and Ghahramani (2012) for a comprehensive discussion.
> In our work, we allow the ensembles to be weighted. Using the uniform weighting reduces to the case above. We have added this to our Discussion section on Line 327.
>
>
> ---
>
> ### References
>
> Belkin, Mikhail, Alexander Rakhlin, and Alexandre B. Tsybakov. "Does data interpolation contradict statistical optimality?." The 22nd International Conference on Artificial Intelligence and Statistics. PMLR, 2019.
>
> Davies, Alex, and Zoubin Ghahramani. "The random forest kernel and other kernels for big data from random partitions." arXiv preprint arXiv:1402.4293 (2014).

---

### Official Review · Reviewer_fiFy · 2022-07-11

**Rating:** 7
**Confidence:** 2
**Soundness:** 4 excellent
**Presentation:** 3 good
**Contribution:** 3 good

**Summary:**

The paper proposes a theoretical analysis of a specific instance of interpolating ensemble methods for classification when the input data lie on a Riemannian manifold. The contributions are two-fold:

- The authors introduce an extension of the Hilbert kernel, denoted the manifold-Hilbert kernel, and show that the classification rule produced by the sign of the Nadaraya-Watson estimator associated with this kernel is both interpolating and consistent.

- They connect ensemble methods to this classification rule through the introduction of weighted random partition, and associated weighted random partition kernels. When a weighted random partition is found to give rise to the manifold-Hilbert kernel, then the corresponding ensemble method is both interpolating and consistent.

This phenomenon is detailed when the manifold is the $d$-dimensional sphere; in this case the proposed random hyperplane arrangement partition kernel corresponds to the manifold-Hilbert kernel associated to the arc-length metric.

**Questions:**

Remarks:
- Links in the pdf are not working.
- Typo line 77: "when" -> we ?
- Figure 1's placement is suboptimal.


**Limitations:**

The authors have adequately addressed the limitations and potential negative societal impact of their work.

**Strengths And Weaknesses:**

Strengths:
- The problem tackled is of interest to the learning community, with an important consistency result.
- The paper is well written and accessible despite the underlying sophisticated mathematics, making it a high quality manuscript.
- Related work in machine learning is well organized and allows to quickly understand the goal of the paper

Weaknesses:

- The heart of the contribution lies in the technical proofs provided by the authors, which I humbly admit to being unable to truly evaluate the novelty of. Essentially, the technical addition of the paper is to extend the results from [19] to the case of the input space being a manifold. While I do not doubt the integrity of the authors, it is hard for me to judge whether the various lemmas (4.1: Riemanniann logarithm, 4.2: change of variable) are groundbreaking in this regard. It would be interesting to have an expert in Riemannian geometry (which I am not) comment on this.

- The theoretical nature of the paper, as well as its technical density may not be the best fit for the NeurIPS community - other learning theory venues such as COLT may be more suited for this.

---

> ### Author Response · Authors · 2022-08-01
> **Technical lemmas and fit at NeurIPS**
>
> We thank the reviewer for the positive comments and the constructive criticisms.
>
> > While I do not doubt the integrity of the authors, it is hard for me to judge whether the various lemmas (4.1: Riemanniann logarithm, 4.2: change of variable) are groundbreaking
> in this regard.
>
> The individual supporting lemmas mentioned by the reviewers are indeed straightforward extensions of well-known results.
> One of our main innovations is putting them together (along our other contributions on weighted random partition kernels) in a nontrivial way to reduce our main result to that of Devroye, Gyorfi, Krzyzak (1998).
>
> These supporting lemmas, while simple, have not appeared in the literature in a way that immediately applies in
> our setting.
> We prove them rigorously for the sake of completeness.
> Our main result also opens the door for future work to simply design kernels on the manifold.
> As such, we believe it is a valuable contribution to rigorously prove these supporting lemmas.
>
> > The theoretical nature of the paper, as well as its technical density may not be the best fit for the NeurIPS
> community - other learning theory venues such as COLT may be more suited for this.
>
> While we agree that COLT would also be an excellent venue, we believe that our work is most suited for the NeurIPS community.
> Differential geometry have been becoming increasingly more explored by the wider machine learning community.
> Our work bridges ideas from differential geometry and learning theory.
> We hope to share our findings so that further connections can be made.
>
> ---
>
> ### References
>
>
> Devroye, Gyorfi, Krzyzak (1998) The Hilbert kernel regression estimate, Journal of Multivariate Analysis.

---

### Official Review · Reviewer_Xx9M · 2022-07-15

**Rating:** 5
**Confidence:** 1
**Soundness:** 3 good
**Presentation:** 3 good
**Contribution:** 3 good

**Summary:**

The paper "Consistent Interpolating Ensembles via the Manifold-Hilbert Kernel" defines the manifold-Hilbert kernel and prove that kernel smoothing regression and classification using it are weakly consistent and hence establishes generalization guarantees in the interpolating regime. For the sphere it is shown that the kernel can be realized as weighted random partition kenrel arising as infinite ensemble of partition-based classifiers, which the authors claim to offer a theoretical basis towards understanding generalization of ensemble of histogram classifiers such as decision trees.

**Questions:**

The authors believe that their work offers a theoretical basis to understand genearlization of for example decision trees. Is there something more concrete than just a believe?

**Limitations:**

Limitations are clearly stated. It is a theoretical analysis under simplifying assumptions and the connection to popular ensemble methods used in practice is still unknown.

**Strengths And Weaknesses:**

I have to say that this is really for away from my expertise, so it was difficult to follow.
- detailed theoretical analysis
- well written, no obvious mistakes I can find
- attempts to improve readability by explaining with words important theory
- 1 result: manifold theory extension of Devroye, which showed that kernel regression with the Hilbert kernel is interpolating and weakly consistent for data with a density and bounded labels.
weaknesses:
- I found it hard to a bit hard to read, since there is a lot of jargon. That is surely my missing background in that area, but maybe that can be helped by putting the relevant background earlier. That helped a bit to understand the motivation of the manuscript
- probably easier to have arabic panels in Fig. 1 and upper left corner instead of somewhere on the lower left.

---

> ### Author Response · Authors · 2022-08-01
> **Clarifying figures and explanations**
>
> We thank the reviewer for the positive comments and the constructive criticisms.
>
> > I found it hard to a bit hard to read, since there is a lot of jargon. That is surely my missing background in that area,
> but maybe that can be helped by putting the relevant background earlier. That helped a bit to understand the
> motivation of the manuscript
>
> We added a new figure (Fig. 1 in the revised version) to make the exposition on the background hopefully more intuitively clear.
>
> > probably easier to have Arabic numeral panels in Fig. 1 and upper left corner instead of somewhere on
> the lower left.
>
> We have incorporated this and modified the figure slightly for better readability. Fig 1 in the first version is now Fig. 2. Thank you for this helpful suggestion.
>
> > The authors believe that their work offers a theoretical basis to understand generalization of, for example, decision trees. Is there something more concrete than just a believe?
>
> The ensemble method we propose is closely related to the so-called PERT (Perfect Random
> Tree) ensembles [Cutler, Zhao (2001)]. Both our method and
> PERT are simplifications of data-dependent ensemble methods (such as Breiman’s random forest).
> These simplifications allow theoretical analysis of consistency to be carried out [Biau, Scornet (2016)]. This expanded explanation has been added to the Discussion section beginning on Line 337 in the revised version.
>
> ---
>
> ### References
>
> Cutler, Adele, and Guohua Zhao. "Pert-perfect random tree ensembles." Computing Science and Statistics 33.4 (2001): 90-4.
>
> Biau, Gérard, and Erwan Scornet. "A random forest guided tour." Test 25.2 (2016): 197-227.

---

### Meta-Review · Area_Chair_XDDr · 2022-08-24

**Recommendation:** Accept
**Confidence:** Certain

**Metareview:**


This paper presents an extension of the Hilbert kernel in Devroye et al (1998) to the Riemannian manifold setting and shows that
kernel smoothing regression is consistent while interpolating the training data on the manifold.

Reviewers generally appreciate the theoretical results presented and agree that this is a well-written paper, despite its technical nature.
The authors did acknowledge the lack of convergence rate, as pointed out by Reviewer w6Vf.
Despite of this limitation, I believe this would make a solid contribution to the theoretical study of interpolating ensemble methods.


Note: Since the original reviewers lack expertise in Riemannian geometry, I asked an expert to provide an additional review (below).
It also contains suggestions for improving the paper. The authors might also want to try to improve their presentation to make it more accessible to the general ML community.

**Added extra review by an expert**

The paper under review extended a result due to Devroye, Gyorfi, and
Krzyzak [DGK98] on the Hilbert Kernel Regression Estimate from the case
$M = \mathbb{R}^n$ to the case M is a complete Riemannian manifold. Its main technique
is the Riemannian logarithm (Theorem 3.2, Lemma 4.1), see also Comment
(2) below. As a result, combining with a remark by Pinelis [Pinelis19],
the author(s) of the paper under review showed that a random partition of
the sphere $S^d$ with a certain weight generates a manifold Hilbert kernel introduced
by the author(s) in Theorem 3.2 and therefore has the interpolating consistent
property (Theorem 5.2, Corollary 5.5). The results are correct,
beautiful and I strongly recommend for publication.

*Comments and suggestions*

(1) l.48: $\alpha$ must belong to $L^1(\beta)$?

(2) l 224: *Alternative proof of Lemma 4.1*. It is known that $M = I_x\cup C_x$
where $I_x$ is diffeomorphic to $\tilde{I}_x \subset T_xM$ and $C_x$ is a closed subset of
M and of the Riemannian measure 0. Now define $\log_x : M \rightarrow T_xM$
as follows: for $m \in I_x$ set $\log_x(m) = \exp^{-1}_x(m)$, for $m \in C_x$ set
$\log_x(m) = 0 \in T_xM$. This map is measurable, since it is continuous
on $I_x$ and it maps the measurable subset $C_x$ of 0-measure to a point.
(So we don't need the measurable selection and the reference Nr
13, which has not been correctly cited, and we don't need supplement
A.1 as well as supplement A.2. (Subsection 2 presents a sufficient
background on Riemannian geometry for this paper in my opinion)

(3) l. 243-247: Propositions 4.2 (i) and (ii): a more general formula in
geometric measure theory, called the area formula, is valid [AT04, p.
44-45]. The author could keep their exposition but should add that
is a simple exposition of a known fact.

*References*

[AT04] Ambrosio, L., Tilli, P.: Topics in Analysis on Metric Spaces. Oxford University
Press, Oxford (2004).

[DGK98] Luc Devroye, Laszlo Gyorfi, and Adam Krzyzak. The Hilbert kernel regression
estimate". In: Journal of Multivariate Analysis 65.2 (1998), pp. 209-227.

[Pinelis19] Iosif Pinelis, Probability of two points being divided by an high-dimensional
hyperplane. MathOverflow. URL:https://mathoverow.net/q/323697 (version:
2019-02-21). 2019.


**Award:**

No

---

### Decision · Program_Chairs · 2022-09-14

Accept